# Perioperative Management and Outcomes after Endovascular Mechanical Thrombectomy in Patients with Submassive (Intermediate-Risk) Pulmonary Embolism: A Retrospective Observational Cohort Study

**DOI:** 10.3390/healthcare12171714

**Published:** 2024-08-27

**Authors:** Michael P. Merren, Mitchell R. Padkins, Hector R. Cajigas, Newton B. Neidert, Arnoley S. Abcejo, Omar Elmadhoun

**Affiliations:** 1Department of Anesthesiology and Perioperative Medicine, Mayo Clinic, Rochester, MN 55902, USA; abcejo.arnoley@mayo.edu (A.S.A.); elmadhoun.omar@mayo.edu (O.E.); 2Department of Cardiovascular Medicine, Mayo Clinic, Rochester, MN 55902, USA; padkins.mitchell@mayo.edu; 3Department of Pulmonary and Critical Care Medicine, Mayo Clinic, Rochester, MN 55902, USA; cajigas.hector@mayo.edu; 4Department of Radiology, Mayo Clinic, Rochester, MN 55902, USA; neidert.newton@mayo.edu

**Keywords:** acute submassive pulmonary embolism, intermediate-risk pulmonary embolism, mechanical thrombectomy, catheter-directed intervention, endovascular therapy, interventional radiology

## Abstract

Pulmonary embolism (PE) embodies a large healthcare burden globally and is the third leading cause of morbidity and mortality worldwide. Submassive (intermediate-risk) PE accounts for 40% of this burden. However, the optimal treatment pathway for this population remains complex and ill-defined. Catheter-directed interventions (CDIs) have shown promise in directly impacting morbidity and mortality while demonstrating a favorable success rate, safety profile, and decreased length of stay (LOS) in the intensive care unit and hospital. This retrospective review included 22 patients (50% female) with submassive PE who underwent mechanical thrombectomy (MT). A total of 45% had a contraindication to thrombolytics, the mean pulmonary embolism severity index was 127, 36% had saddle PE, the average decrease in mean pulmonary artery pressure (PAP) was 7.2 mmHg following MT, the average LOS was 6.9 days, the 30-day mortality rate was 9%, the major adverse event (MAE) rate was 9%, and the readmission rate was 13.6%. A total of 82% had successful removal of thrombus during MT with no major bleeding complications, intracranial hemorrhage events, or device-related deaths. Acknowledging the limitation of our small sample size, our data indicate that MT in the intermediate-high-risk submassive pulmonary embolism (PE) cohort resulted in a decreased hospital length of stay (LOS) and in-hospital mortality compared to standard anticoagulation therapy alone.

## 1. Introduction

Pulmonary embolism (PE) presents a global healthcare burden with over one million individuals diagnosed annually [1]. It ranks as the third leading cause of death, trailing only myocardial infarction and stroke, both in the United States and around the globe [2]. The incidence of PE and associated hospitalizations is also rising, imposing clinical and financial strain on the healthcare system [3,4]. Despite our experience in studying this disease, the outcomes remain suboptimal with high mortality, recurrent disease, and increased rehospitalizations with a recent retrospective review showing the 1-year rehospitalization rate to be 48.6% in 55,000 patients hospitalized for PE [5]. These issues are not solely attributed to patient comorbidities but are also closely related to the management strategies undertaken.

Currently the American Heart Association (AHA) classifies PE into three mortality risk groups: low-risk, submassive, and massive [6]. The European Society of Cardiology (ESC) classifies early mortality risk into low-, intermediate- (with a subcategory of intermediate-low- and intermediate-high-), and high-risk [7]. Submassive and intermediate-risk PE represent a population characterized by hemodynamic stability but evidence of RV injury or dysfunction, identified through biomarkers or imaging. Submassive PE accounts for 40% of PEs, with a 30-day mortality rate ranging from 5 to 25% [8]. While there is a consensus regarding managing low-risk and massive PE, determining the optimal approach for submassive PE patients remains in flux. Despite new interventions, management remains conservative, mainly relying on anticoagulation (AC) alone. This approach may prove insufficient for patients exhibiting signs of impending hemodynamic compromise, in which AHA and ESC guidelines suggest an escalation of therapy to the use of thrombolytics [9,10]. However, the use of thrombolytics comes at the expense of an increased risk of devastating complications without definitive evidence for mortality or functional outcome improvement [11].

Over the past few decades, various percutaneous catheter-directed interventions (CDIs) have emerged for treating PE [12]. Initially introduced for high-risk PE cases with contraindications for thrombolysis, the utilization of CDIs is expanding in submassive PE patients as well. Despite several recent and ongoing studies investigating clinical and safety outcomes [13,14,15], widespread adoption of these interventions is not yet recommended. Potential benefits of CDIs include a measurable hemodynamic response in the interventional suite, quicker improvement in hemodynamics, and reduced bleeding risk compared to systemic thrombolytic therapy. However, risks may include vascular or cardiac injury and the potential need for inducing general anesthesia in high-risk patients [16].

Our investigation assessed patients with intermediate-risk or submassive PE treated utilizing endovascular mechanical thrombectomy (MT) techniques in interventional radiology suites. Using data collected from patients admitted to a large tertiary hospital enterprise, we aimed to ascertain the acute success, safety, and 30-day rehospitalization rates in these individuals. Our goal is to contribute to the scarce literature on this subject and provide insights to help guide decision making at the bedside.

## 2. Methods

This retrospective observational cohort study was approved by the Mayo Clinic Institutional Review Board (IRB 23-001636-02). Retrospective data were collected by investigators using the Mayo Clinic Informatics for Integrating Biology in the Bedside (i2b2) database with ICD 10 codes along with a diagnosis of pulmonary embolism. The ICD 10 codes included were 37187, 37188, 37184, 37185, and 37186. Patients presenting with intermediate-risk or submassive PE (which we will refer to as just submassive PE from this point forward) who underwent percutaneous mechanical thrombectomy were identified between 1 January 2017 and 1 September 2023, at a single, quaternary care institution. Intermediate-risk PE was defined by the 2019 ESC guidelines as a combination of (1) acute PE on computed tomographic pulmonary angiography (CTPA), (2) calculated pulmonary embolism severity (PESI) class III-V, or simplified PESI greater or equal to a score of one, and (3) either evidence of right ventricular (RV) strain on CTPA or transthoracic echocardiogram (TTE) or an elevated cardiac troponin I or T level and (4) with hemodynamic stability [7]. Hemodynamic instability was defined as any one of the following clinical presentations: cardiac arrest, obstructive shock (systolic BP <90 mmHg or vasopressors required to achieve a systolic BP >90 despite an adequate filling status, in combination with end-organ hypoperfusion), or persistent hypotension (systolic BP <90 or a systolic drop greater or equal to 40 mmHg for >15 min, not caused by new-onset arrhythmia, hypovolemia, or sepsis) [7]. The American Heart Association classification of submassive PE was defined clinically as acute PE on CTPA along with systolic blood pressure > 90 mmHg and RV dysfunction or myocardial necrosis defined by the following: RV dilation (apical 4-chamber RV diameter divided by left ventricular (LV) diameter >0.9) or RV systolic dysfunction on echocardiography or RV dilation (4-chamber RV diameter divided by LV diameter >0.9) on CT or elevation in BNP (>90 pg/mL), or N-terminal pro-BNP (>500 pg/mL) or EKG changes (new complete or incomplete right bundle branch block, anteroseptal ST-segment elevation or depression, or anteroseptal T-wave inversion) or elevation in troponin I (>0.4 mg/mL) or troponin T (>0.1 ng/mL) [6].

The primary aim of our research was to describe the in-hospital and 30-day mortality rate, length of hospital stay, procedure-related major adverse events and major bleeding complications among patients who underwent MT for submassive PE at our institution. Our secondary aim was to investigate patient characteristics associated with increased risk of death or complications, 30-day readmission rate, and 30-day recurrent PE or DVT episodes. 

Catheter-directed MT was performed with the FlowTriever Retrieval/Aspiration System (Inari Medical, Irvine, CA, USA). This thrombectomy system includes 24 Fr, 20 Fr, and 16 Fr aspiration catheters that can be used independently or in telescoping fashion with each other to aspirate thrombus from the pulmonary arteries. The system also includes catheters of braided nitinol disks that can be used to disrupt thrombus in conjunction with the aspiration catheters mechanically.

Ultrasound-guided percutaneous access for MT can be obtained via the common femoral or internal jugular veins, but femoral venous access is typically favored. A catheter is advanced through the right heart chambers into the central pulmonary arteries, and pulmonary arterial pressures are obtained. Pulmonary angiography is then performed, and contrast injection is tailored to the pulmonary pressures to avoid cardiovascular complications. Once the thromboembolic burden is delineated, systemic procedural anticoagulation is achieved, and the thrombectomy catheters are advanced into the pulmonary arteries over guidewires. Aspiration thrombectomy and, if needed, mechanical thrombectomy are performed to remove as much thrombus burden as possible from the right and left main pulmonary arteries and lobar branches. Pulmonary angiography is repeated to reassess the patency of the pulmonary arteries after thrombectomy. Pulmonary artery pressures are repeated to assess hemodynamic response to thrombectomy.

Patients were excluded from the analysis if they were less than 18 years old, considered part of a vulnerable population (as defined by FDA regulations), or patients whose medical records did not include research authorization, who received medical therapy alone, who were diagnosed with less than submassive PE, or who were diagnosed with a massive or high-risk PE.

Data analysis consisted of patient demographics including age, sex, and body mass index (BMI). Further patient characteristics included concurrent deep vein thrombosis (DVT), prior history of PE, prior history of DVT, surgery within two weeks of presentation, contraindications to thrombolytics per AHA guidelines [6], active cancer, and calculated PESI score. Multiple authors participated in data gathering along with reviewing each patient’s chart to ensure data accuracy and completeness.

Vital sign data were collected including initial presenting heart rate, systolic blood pressure, and oxygen saturation. Laboratory pre-procedure data were obtained including cardiac biomarkers such as NT-pro B natriuretic peptide assay (BNP) (Roche Diagnostics, Indianapolis, Indiana) and cardiac troponin concentration (cTnT) Elecsys Troponin T Gen 5 STAT assay (Roche Diagnostics, Indianapolis, Indiana) along with imaging evidence of right heart strain on either CTPA or TTE and the anatomic location of the PE. TTE parameters were also obtained pre- and post-procedure including evidence of right heart strain, right ventricular (RV) diameter to left ventricular (LV) diameter ratio, LV function, RV systolic pressure (RVSP), tricuspid annular plane systolic excursion (TAPSE), and TAPSE/RVSP ratio. Procedure measurements of mean pulmonary artery pressure before and after MT completion were collected.

Procedure-related major adverse events (MAE) were defined as any of the following: device-related death, major bleeding, treatment-related clinical deterioration, and device-related severe adverse events which include a composite of clinical deterioration, pulmonary vascular injury, and cardiac injury were collected. Outcome-related information including average length of hospital stay (LOS), in-hospital death, death within 30 days, recurrent PE or DVT within 30 days, and readmission within 30 days of undergoing MT was collected and analyzed as the primary outcome.

Categorical data were compared via Chi-squared analysis. Continuous data were compared via an analysis of variance with post hoc unpaired Student’s *t*-tests, where appropriate. A 2-tailed *p*-value < 0.05, corrected for multiple comparisons, was considered statistically significant. Specifically, when data among the four groups stratified based on injury type were compared via post hoc unpaired Student’s *t*-tests, a *p*-value of <0.008 was considered statistically significant.

## 3. Results

### 3.1. Patient Characteristics

Over the study period, 3852 patients were screened positive for a diagnosis of acute PE and/or underwent a mechanical thrombectomy. We excluded 3817 patients for not meeting our inclusion criteria. A total of 35 patients remained, and another 13 were excluded as they met massive PE criteria, leaving us with 22 patients that met our inclusion criteria, as seen in Figure 1. Patient characteristics are listed in Table 1, and statistics are listed in Table 2. The mean age was 65.8 (30 to 88) with 11 female patients and a mean BMI of 31.5 (19.4–42.8). Fourteen patients (64%) had concurrent DVT, with five patients (23%) having had surgery within two weeks of developing a submassive PE. A total of 11 patients (50%) had a contraindication to thrombolytics, with one patient receiving thrombolytics in the emergency room before MT. Five patients (23%) had active cancer at the time of diagnosis of PE. Mean values for the pulmonary embolism severity index (PESI) were 127 (44–231), a cTnT level of 79.6 (6–313) ng/L (normal <15 mg/L), and a BNP level of 3117 (225–15,437) pg/mL (normal <300 pg/mL). Eight patients (36%) had a radiographically described saddle PE. The mean pre-MT mPAP was 30 (19–50) mmHg, and the post-MT mPAP was 22.8 (15–37) mmHg. The average hospital LOS was 6.9 (1–21) days. In this group, there were two patients (9%) with recurrent PE, three (13.6%) requiring readmission, and two (9%) who died within 30 days of undergoing MT. Of the two patients who died, one died during the MT procedure after suffering a cardiac arrest. The second patient had widely metastatic colon cancer and had no issues during their hospital admission or MT procedure and were subsequently discharged to a skilled nursing facility, where they died ten days after hospital discharge. For a more detailed description of the two patients who died, see Appendix A.

### 3.2. Echocardiographic Data

Transthoracic echocardiograms (TTE) were performed in 11 patients (50%) before MT, as shown in Table 3. The mean values for the right ventricular to left ventricular ratio were 1.26 (0.73–2.0), mean values for TAPSE were 14.7 (10–24) mm, RVSP was 43.9 (15–76) mmHg, and TAPSE/RVSP ratio was 0.43 (0.2–0.88) mm/mmHg.

### 3.3. Mechanical Thrombectomy Device-Related Data

MT was performed on 22 patients, as shown in Table 4. A total of 18 patients (82%) were able to have thrombus successfully removed during the procedure, with no patients suffering from either significant bleeding or device-related death. Two patients (9%) had a major adverse event (MAE) within the first 48 h. The two MAEs included one patient with a guidewire induced left pulmonary artery perforation causing hemoptysis that self-resolved without the need for blood product transfusion or intubation. The other patient suffered a hypoxemic respiratory arrest and died during the procedure in which the thrombus was not able to be removed.

## 4. Discussion

In this single-center, retrospective observational cohort study of patients undergoing MT via the FlowTriever Retrieval/Aspiration System (Inari Medical, Irvine, CA, USA), we describe a 91% 30-day survival rate in patients with submassive PE with an average LOS of 6.9 days, a 9% recurrent PE rate, a 13.6% readmission rate, and a 9% device-related major adverse event rate.

### 4.1. Predicting Mortality

RV dysfunction and elevated cardiac biomarkers have been shown to have predictive value for 30-day clinical deterioration with acute PE. In a large meta-analysis of more than 13,000 patients, the RV/LV ratio >1.0 measured on transverse CT angiography sections had a 2.5 increased risk for all-cause mortality and a 5-fold increase in PE-related mortality [17]. Two echocardiographic measurements, an RV/LV ratio greater than 1.0 in the apical four-chamber view and a TAPSE <15 mm, are associated with increased hospital and 30-day mortality as well as the need for rescue thrombolytic therapy [18,19,20]. From a biomarker measurement perspective, the presence of elevated cardiac troponins in a meta-analysis of 1985 patients was shown to be associated with an increased risk of clinical decompensation, in-hospital mortality, and death, thus adding prognostic value to patients with submassive PE [21,22]. Moreover, in a meta-analysis of 13 studies by Klok et al., an increase in BNP level highly correlated with RV dysfunction on TTE and increased the chances of early death by 10% and clinical deterioration by 23% [23]. In our study group, 21 patients (95.5%) had an elevation in both cTnT and BNP, and 20 of those had concurrent evidence of right heart strain from a CT scan or TTE. Based on this, our cohort of patients represents one of the highest-risk submassive PE groups for further decompensation and 30-day mortality.

### 4.2. Comparing Mortality Outcomes

Submassive or intermediate PE patients represent a group that is hemodynamically stable but have evidence of right heart strain or elevation in cardiac biomarkers and make up 40% of the PE population with a mortality rate approaching 25% [8]. The 2011 AHA and 2019 ESC guideline recommendations for the treatment of submassive (AHA) and intermediate-risk (ESC) PE are clear: Systemic AC should be started immediately if there are no contraindications [6,7]. However, the decision to use other available therapies is left to the providing team’s discretion.

The first sizable mechanical thrombectomy study, the FLARE trial, evaluated 106 patients with submassive PE undergoing MT with the FlowTreiver system (Inari Medical, Irvine, CA, USA). This study reported no in-hospital deaths and a 1% 30-day mortality rate. There was a 3.8% major adverse event within 48 h, a 1% major bleeding incident, and no ICH events [24]. Subsequently, utilizing the FlowTreiver system, the FLASH study evaluated 800 patients, with 76.7% having intermediate-high-risk and 7.9% with high-risk PE undergoing MT. The study did not perform subgroup analysis breaking out the different risk categories in its results, but they reported an in-hospital mortality rate of 0.3% and a 30-day mortality rate of 0.8%. The study found 1.8% MAE with 1.5% major bleeding incidents without ICH events [15]. A third mechanical thrombectomy study, the EXTRACT-PE, utilized the Indigo Aspiration MT system (Penumbra, Alameda, CA, USA) and evaluated 118 patients with submassive PE. They reported an in-hospital and 30-day mortality rate of 0.8% and 2.5%, respectively, which was similar to the other mechanical thrombectomy mortality rates. The MAE and major bleeding rates were both 1.7% with one reported patient device-related death and no reported ICH events [25].

Looking retrospectively at studies that included submassive PE patients that were only anticoagulated as a comparison group, the PEITHO trial compared a total of 905 patients receiving either systemic thrombolytics versus AC alone and found that in the AC treatment arm an in-hospital mortality rate of 1.8%, 30-day mortality rate of 3.2%, and major bleeding rate of 2.4% [9]. In a large meta-analysis of 7918 patients looking at catheter-directed thrombolytics compared to AC alone, they found in the AC alone arm an in-hospital mortality rate of 6.4%, 30-day mortality rate of 10% and major bleeding rate of 2.8% [26]. In a study by Buckley et al. comparing MT to AC alone in 58 patients, they found an in-hospital mortality rate of 23% and major bleeding rate of 3.3% in the AC-alone group [27].

In our current study of 22 patients, we found an in-hospital mortality and 30-day mortality rate of 4.5% and 9%, respectively, which are higher those reported in the FLARE, FLASH, and EXTRACT-PE studies [15,24,25]. We also found an MAE rate of 9% and did not find any major bleeding events. This is likely due to our higher-risk patient cohort and smaller sample size. In our study cohort, there were 3 patients (13.6%) in the intermediate-low-risk and 19 patients (86.4%) in the intermediate high-risk submassive PE categories based on PESI scoring alone. Another possible contributing factor was that 25% of our study group had active cancer. Cancer-related PE prognosis is poor, with mortality rates alone roughly 24.2% at 30 days and an increased in-hospital mortality of 24.8% vs. 6.5% without cancer-related PE [28]. The FLARE trial excluded active cancer patients, while the EXTRACT-PE study included 22% of their patients with any history of cancer but excluded those with active cancer undergoing chemotherapy, and the FLASH study included only 8% with active cancer.

### 4.3. ICU and Hospital LOS

The FLARE study reported a mean LOS of 4.1 ± 3.5, and the FLASH study reported three hospital overnights post-procedure [15,24]. The FLARE and EXTRACT-PE studies reported mean ICU LOS at 1.5 and 1.0 days, respectively, following MT [24,25]. In a study by Khazi et., 41 MT patients and 82 patients in the AC group were evaluated. They found there was no significant difference in ICU LOS. However, there was a significant difference in hospital LOS, revealing 5.37 ± 3.93 days in the MT group and 9.53 ± 4.49 in the AC group. They also found a significant decrease in the 30-day readmission rate: 5.26% in the MT group vs. 26.4% in the AC group [29]. In another study by Buckley et al., they evaluated 28 MT patients and 30 AC patients with submassive PE; they found a significant difference in ICU LOS, reporting 2.1 ± 1.2 days for MT and 6.1 ± 8.6 days for AC but similar hospital LOS of 7.7 ± 6.9 days for MT and 6.8 ± 6.9 days for AC [27].

In our group of MT patients, we found that the mean LOS was 6.9 days. This is statistically higher than the FLARE and FLASH studies but similar to the Buckley and Khazi et al. studies [15,24,27,29]. Our MT patients also experienced a lower mean LOS compared to the AC group in the Ismayl et al. study [26]. A likely explanation for this is that our group of patients had a comparatively elevated risk for mortality based on their high PESI scores and a combination of right heart strain plus elevated cardiac biomarkers. As previously mentioned, 25% of our study group had active cancer, which can add additional LOS to their hospitalization since a new diagnosis of PE could indicate the progression or a more detailed in-hospital cancer workup. In a recent review by Welker et al., they found that the development of PE in cancer patients represents a significant increase in mortality. They also found increased healthcare costs associated with higher rates of readmissions that may be reduced using MT in this population compared to AC alone [30]. We conclude that we may have a different patient population when comparing our patient cohort LOS to the FLARE and FLASH studies. Lastly, our group likely carries with it a real-world selection bias. These patients were evaluated and identified as having a higher risk of mortality and were thus referred for MT in the first place.

## 5. Limitations

First, the sample size is small, from a single center, and this study is retrospective in design. These factors alone can lead to selection bias and confounders, making the generalizability limited. Secondly, we had a minimal number of follow up TTEs at 48 h to evaluate for reductions in RV strain or reductions in RV size. We could not report any outcome data from our study outside of hospital LOS and death within 30 days of MT. This limited our discussion regarding any clinical change in echocardiographic data. Thirdly, our cohort of patients represented higher-risk patients than any other study. This represents a selection bias in our group; however, we reinforced the safety profile of MT in the intermediate high-risk submassive PE population. Our study did not report economic implications, and we cannot demonstrate the financial impact of MT on healthcare in this group of patients with submassive PE. Furthermore, our study did not evaluate long-term outcomes following MT.

## 6. Conclusions

In our small cohort of patients with intermediate-risk submassive PE, there was a benefit with in-hospital mortality and hospital LOS without any major bleeding events. Our study has added to the growing body of evidence revealing favorable risk/benefit profile of the use of mechanical thrombectomy in the submassive PE population. Future large prospective randomized controlled studies are needed to confirm our findings.

## Figures and Tables

**Figure 1 healthcare-12-01714-f001:**
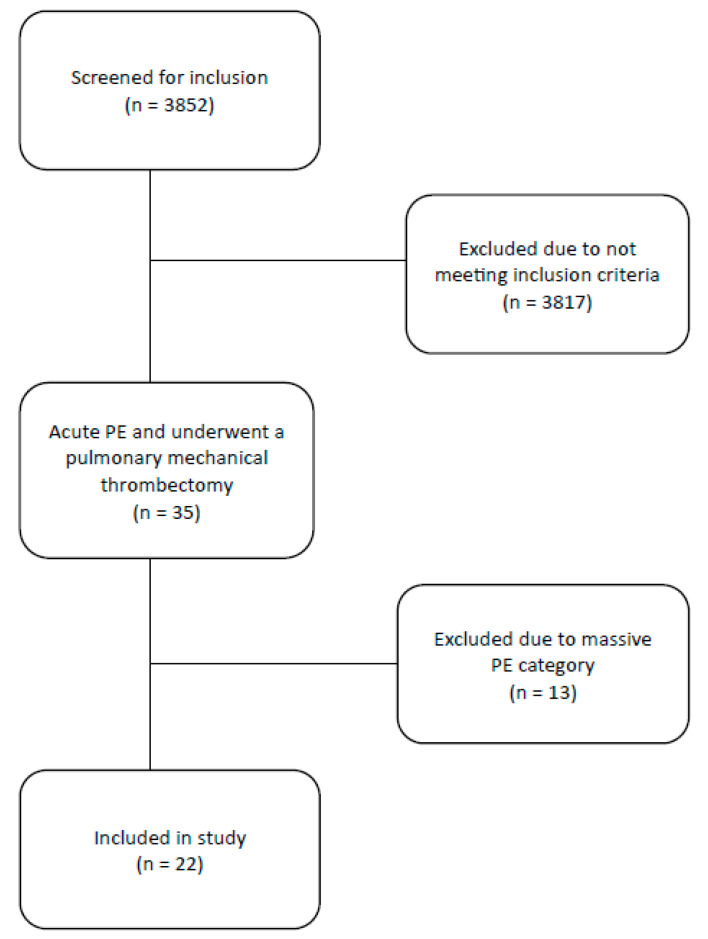
A flow diagram for the inclusion of acute submassive pulmonary embolism patients undergoing mechanical thrombectomy.

**Table 1 healthcare-12-01714-t001:** Patient characteristics.

Patient	Age	Sex	BMI	Concurrent DVT	h/o PE	h/o DVT	Surgery within 2 weeks of PE	Contraindication to Thrombolytics	Active Cancer	RV Dysfunction on ECHO or CT	PESI Score
1	45	female	42.8	yes	no	no	no	no	no	yes	116
2	80	female	20.3	yes	no	no	no	no	no	yes	180
3	34	male	34.3	yes	no	no	no	no	no	no	44
4	88	female	26.9	yes	no	no	no	yes	no	yes	147
5	68	male	23.1	no	no	no	no	no	no	no	88
6	71	male	33.5	no	no	no	yes	yes	no	yes	181
7	65	female	29.4	yes	no	no	yes	yes	no	yes	105
8	68	male	34	yes	no	no	no	no	no	yes	168
9	63	male	28.9	yes	no	no	no	no	yes	yes	123
10	83	female	28	yes	no	no	no	yes	yes	yes	143
11	73	female	26.5	yes	no	no	no	yes	yes	yes	153
12	74	male	27.9	yes	yes	no	no	no	yes	yes	134
13	75	female	39.8	no	no	no	no	yes	no	yes	115
14	30	male	30.6	yes	no	no	yes	yes	yes	yes	90
15	63	male	32.1	yes	no	no	yes	yes	no	yes	163
16	69	male	26.5	yes	no	no	no	yes	no	yes	129
17	74	male	29.3	yes	yes	no	yes	no	no	yes	134
18	61	female	41.2	not reported	no	no	no	no	no	yes	231
19	60	female	42.3	no	yes	yes	no	yes	no	yes	61
20	68	female	38.6	no	yes	yes	no	no	no	yes	81
21	64	female	19.4	yes	no	no	no	no	no	yes	94
22	72	male	38.6	not reported	no	no	no	yes	no	yes	122
Patient	Troponin level (ng/L)	BNP level (pg/mL)	Initial presenting systolic BP (mmHg)	Location of PE	Pre MT mPAP (mmHg)	Post MT mPAP (mmHg)	Death within 30 days	Length of stay (days)	Recurrent PE in 30 days	Readmission within 30 days	
1	19	528	131	right main	37	25	no	4	yes	no	
2	248	304	114	saddle, right and left main	26	17	no	4	no	no	
3	6	317	143	bilateral main	22	15	no	10	no	no	
4	210	538	141	saddle with occlusive right main	21	15	no	10	no	no	
5	11	970	104	left main	33	33	no	8	no	no	
6	44	not reported	124	saddle, right and left main	24	19	no	11	no	no	
7	92	not reported	135	bilateral segmental in all 3 branches of right lung	34	28	no	7	no	no	
8	59	not reported	89	saddle and bilateral segmental branches	28	23	no	3	no	no	
9	120	5458	145	saddle and bilateral segmental branches	29	20	no	6	no	yes	
10	44	8074	134	right main and left segmental	not reported	not reported	no	4	no	no	
11	32	1047	118	saddle and bilateral segmental branches	19	not reported	yes	10	no	no	
12	86	5532	114	bilateral main PA and segmental	not reported	not reported	no	3	no	no	
13	55	7749	147	bilateral segmental branches	not reported	not reported	no	7	no	yes	
14	16	225	88	bilateral segmental branches	not reported	not reported	no	7	yes	yes	
15	86	not reported	109	right main and bilateral segmental branches	26	not reported	no	21	no	no	
16	53	692	98	saddle and bilateral segmental branches	not reported	not reported	no	7	no	no	
17	313	953	97	bilateral main PA and segmental	35	not reported	no	5	no	no	
18	114	2974	107	saddle and bilateral segmental branches	46	not reported	yes	1	no	no	
19	10	296	156	distal right main	25	25	no	5	no	no	
20	22	not reported	120	bilateral main and lobar branches	27	17	no	11	no	no	
21	53	15,437	105	left main and segmental branches	50	37	no	5	no	no	
22	59	1900	128	bilateral distal main, lobar, and segmental branches	not reported	not reported	no	3	no	no	

**Table 2 healthcare-12-01714-t002:** Patient characteristic statistical analysis.

Characteristic	Number of Patients	Mean	Standard Deviation	Standard Error	95% Confidence Interval
PESI score	22	127.4	43.1	9.2	108.2
Troponin level (ng/L)	22	79.6	80.6	17.2	43.9
BNP level (pg/mL)	17	3117.3	4149.6	1006.4	983.8
Initial presenting systolic BP	22	120.3	19.6	4.2	111.6
Pre-thrombectomy mean PAP (mmHg)	16	30.1	8.7	2.2	25.5
Post-thrombectomy mean PAP (mmHg)	12	22.8	7.1	2.0	18.3
RV/LV ratio pre-thrombectomy	11	1.3	0.4	0.1	1.0
Pre-thrombectomy TAPSE	11	14.7	4.7	1.4	11.6
Age	22	65.8	14.1	3.0	59.6
BMI	22	31.5	6.8	1.5	28.5

**Table 3 healthcare-12-01714-t003:** Transthoracic echocardiography data.

Patient	RV/LV Ratio Pre-Thrombectomy	Pre-Thrombectomy TAPSE (mm)	Pre-Thrombectomy RVSP on ECHO (mmHg)	Pre-Thrombectomy RV to PA Coupling (TAPSE/RVSP Ratio in mm/mmHg)
1	not reported	not reported	not reported	not reported
2	2	12	40	0.3
3	0.88	23	26	0.88
4	1.1	10	49	0.2
5	not reported	not reported	not reported	not reported
6	1.8	not reported	47	not reported
7	1.3	16	50	0.32
8	not reported	not reported	not reported	not reported
9	not reported	14	36	0.39
10	not reported	not reported	44	not reported
11	1.06	24	52	0.46
12	1.17	11	15	0.73
13	not reported	not reported	76	not reported
14	0.73	not reported	not reported	not reported
15	not reported	not reported	53	not reported
16	1.4	13	29	0.45
17	not reported	11	23	0.48
18	not reported	not reported	not reported	not reported
19	not reported	not reported	44	not reported
20	1.09	13	59	0.22
21	1.3	15	59	0.25
22	not reported	not reported	not reported	not reported

**Table 4 healthcare-12-01714-t004:** Mechanical thrombectomy device-related data.

Patient	Able to Remove Clot during Procedure	Device-Related Death	Major Bleeding	Major Adverse Event in First 48 h after Procedure
1	yes	no	no	no
2	yes	no	no	no
3	yes	no	no	no
4	yes	no	no	no
5	no	no	no	no
6	yes	no	no	no
7	no	no	no	no
8	yes	no	no	Yes, there was a perforation of the left pulmonary artery with the guidewire causing hemoptysis without any need for transfusion or intubation.
9	yes	no	no	no
10	yes	no	no	no
11	yes	no	no	no
12	yes	no	no	no
13	yes	no	no	no
14	yes	no	no	no
15	yes	no	no	no
16	yes	no	no	no
17	yes	no	no	no
18	no	no	no	Yes, during the procedure, the patient had a respiratory arrest with hypoxemia requiring intubation and then developed PEA arrest without achieving ROSC. The proceduralist was unable to remove any thrombus during the procedure.
19	yes	no	no	no
20	yes	no	no	no
21	no	no	no	no
22	yes	no	no	no

## Data Availability

Data can be obtained by emailing the primary author at merren.michael@mayo.edu.

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
