# Peer review of "Perioperative Management and Outcomes after Endovascular Mechanical Thrombectomy in Patients with Submassive (Intermediate-Risk) Pulmonary Embolism: A Retrospective Observational Cohort Study"

_healthcare, 2024, doi:10.3390/healthcare12171714_

Round 1

Reviewer 1 Report

Comments and Suggestions for Authors

Reviewer Comments

The manuscript (healthcare-3124737) submitted by Merren et al. deals with the retrospective study about Perioperative management and outcomes after endovascular mechanical thrombectomy in patients with submassive pulmonary embolism. Apart from correcting formatting issues, such as inconsistent use of abbreviations and missing citations for some statements, the authors may look into the following comments which might help them to improve the manuscript:

1.     The introduction should be more focused narrative. Rather than discussing multiple aspects of PE without a clear transition between different points, a more streamlined flow would improve readability.

2.     The interpretation the findings in the context of existing literature is neeeded. Comparing the study results with previous studies would enhance the understanding of the study's impact.

3.     The authors should emphasize the clinical significance of the study findings, particularly in guiding treatment decisions for submassive PE.

4.     The limitations of the study should be discussed, including the retrospective design and potential biases, with regards to small sample size and its impact on generalizability.

5.     There is no mention of how data accuracy and completeness were ensured, as this is a critical aspect that needs to be addressed to validate the reliability of the study findings.

6.     The discussion on risk groups according to AHA and ESC guidelines could be more detailed.

7.     The primary and secondary outcome measures are not clearly differentiated.

8.     Suggest areas for future research, especially prospective studies and larger trials to confirm these findings.

9.     The conclusion needs to provide a clearer and more concise summary of the study's key findings. The current conclusion is somewhat vague and does not fully encapsulate the study's results.

Comments on the Quality of English Language

Moderate editing of English language required

Author Response

Comment 1: The introduction should be more focused narrative. Rather than discussing multiple aspects of PE without a clear transition between different points, a more streamlined flow would improve readability. 

Response 1: We have reviewed the introduction and feel that it follows a logical order of introducing the topic. The first paragraph discusses the burden of disease, the second paragraph discusses the classification of the disease, the third discusses treatment of the disease, and the last paragraph discusses our study’s treatment to be evaluated. 

Comment 2: The interpretation the findings in the context of existing literature is needed. Comparing the study results with previous studies would enhance the understanding of the study's impact. 

Response 2: Under the discussion section (section 4), subsection 4.2 discusses the current studies published on mechanical thrombectomy use in submassive PE and the last paragraph compares our data to those larger studies. However, we did not add in our major adverse events or major bleeding events in this section. I have added it to the manuscript in the last paragraph of section 4.2 

Comment 3: The authors should emphasize the clinical significance of the study findings, particularly in guiding treatment decisions for submassive PE. 

Response 3: We agree with this and have changed our conclusion to read:  “In our small cohort of patients with intermediate risk submassive PE, there was a benefit with in-hospital mortality and hospital LOS without any major bleeding events. Our study has added to the growing body of evidence revealing favorable risk/benefit profile of the use of mechanical thrombectomy in the submassive PE population. Future large prospective randomized controlled studies are needed to confirm our findings. 

Comment 4: The limitations of the study should be discussed, including the retrospective design and potential biases, with regards to small sample size and its impact on generalizability. 

Response 4: We agree with the above and have changed the limitations section in our manuscript accordingly. 

Comment 5: There is no mention of how data accuracy and completeness were ensured, as this is a critical aspect that needs to be addressed to validate the reliability of the study findings. 

Response 5: Thanks for pointing this out, we have added the following: “Multiple authors participated in data gathering along with reviewing each patient’s chart to ensure data accuracy and completeness.” to the 5 paragraph of the methods section. 

Comment 6: The discussion on risk groups according to AHA and ESC guidelines could be more detailed. 

Response 6: This is discussed in detail in the introduction section, 2nd paragraph. We have added a sentence to the beginning of section 4.2 to read: “Submassive or intermediate PE patients represent a group that is hemodynamically stable but have evidence of right heart strain or elevation of cardiac biomarkers and make up 40% of the PE population with a mortality rate approaching 25%” 

Comment 7: The primary and secondary outcome measures are not clearly differentiated. 

Response 7: we agree with the reviewer and added a paragraph to the methods section that reads: “The primary aim of our research was to describe the in-hospital and 30-day mortality rate, length of hospital stay, and procedure related major adverse events among patients that underwent MT for submassive PE at our institution. Our secondary aim was to investigate patient characteristics associated with increased risk of death or complications, 30-day readmission rate, and 30-day recurrent PE or DVT episodes. We also sought to determine the procedure related major adverse events, major bleeding complications, length of stay, 30-day readmission rate, and 30-day   

Comment 8: Suggest areas for future research, especially prospective studies and larger trials to confirm these findings. 

Response 8: we agree with the reviewer and added this to our conclusion paragraph 

Comment 9: The conclusion needs to provide a clearer and more concise summary of the study's key findings. The current conclusion is somewhat vague and does not fully encapsulate the study's results. 

Response 9: we agree with the reviewer and have re-written the conclusion paragraph. In our small cohort of patients with intermediate risk submassive PE, there was a benefit with in-hospital mortality and hospital LOS without any major bleeding events. Our study has added to the growing body of evidence revealing favorable risk/benefit profile of the use of mechanical thrombectomy in the submassive PE population. Future large prospective randomized controlled studies are needed to confirm our findings.” 

Reviewer 2 Report

Comments and Suggestions for Authors

This study provides a valuable contribution to the optimization of PE management in clinical practice, especially given the high mortality associated with the disease. The retrospective cohort study design is appropriate, and the results are straightforward. However, there are a few minor issues that could be improved:

  1. Tables 1, 2, 3, and 4: These tables, which present data for each participant, are very informative for readers. However, it would be better to present this information in a summary table with descriptive analysis to show the demographics and clinical characteristics of the studied group who underwent MT. This is important for clinical reference.

  2. Table 4: Given the small sample size, it would be better to provide a summary table to show each endpoint and the outcome. This is the main result, which needs to be shown clearly to the reader.

  3. Control Group: If possible, add a control group of patients who underwent conventional anticoagulation management. This group could  be selected from the dataset with matched gender, age, and submassive PE diagnosis using equivalent diagnostic standards. Can do the comparison of outcomes between the study group and the control group and show the results in the summary table.

Author Response

Comment 1: Tables 1, 2, 3, and 4: These tables, which present data for each participant, are very informative for readers. However, it would be better to present this information in a summary table with descriptive analysis to show the demographics and clinical characteristics of the studied group who underwent MT. This is important for clinical reference. 

Response 1: Thank you for your suggestion, however we feel that our tables efficiently display the data for comparison very well. 

Comment 2: Table 4: Given the small sample size, it would be better to provide a summary table to show each endpoint and the outcome. This is the main result, which needs to be shown clearly to the reader. 

Response 2: we agree with the reviewer and have revised Table 4 in the document in order to give more details on the two patients that had major adverse events in the first 48 hours after the procedure. 

Comment 3: Control Group: If possible, add a control group of patients who underwent conventional anticoagulation management. This group could be selected from the dataset with matched gender, age, and submassive PE diagnosis using equivalent diagnostic standards. Can do the comparison of outcomes between the study group and the control group and show the results in the summary table. 

Response 3: We had thought about using patients from our own institution initially, however we wanted to limit as much bias at a single institution as possible. With that in mind we decided to use reference 29 (large meta-analysis with 7,918 patients treated with anticoagulation alone) as our comparative group instead.  

Reviewer 3 Report

Comments and Suggestions for Authors

Limited Sample Size: A small patient cohort may restrict the generalizability of the findings. A larger group would yield more robust data and enhance the reliability of the results.

Lack of Long-term Follow-up: The study might emphasize short-term outcomes like 30-day mortality and readmission rates, overlooking long-term impacts such as quality of life and functional status, which are essential for a comprehensive assessment of the intervention.

Absence of Control Group: Without a control group for comparison, it may be challenging to determine the effectiveness of mechanical thrombectomy compared to standard treatments like anticoagulation alone.

Potential Selection Bias: The retrospective design of the study might introduce selection bias, as the patients included may not accurately represent the broader population with submassive PE, potentially compromising the validity of the findings.

Author Response

Comment 1: Limited Sample Size: A small patient cohort may restrict the generalizability of the findings. A larger group would yield more robust data and enhance the reliability of the results. 

Response 1: We agree with the reviewer that the small cohort does limit generalizability, but it does add to the growing body of evidence with the mechanical thrombectomy safety and impact on length of stay. 

Comment 2: Lack of Long-term Follow-up: The study might emphasize short-term outcomes like 30-day mortality and readmission rates, overlooking long-term impacts such as quality of life and functional status, which are essential for a comprehensive assessment of the intervention. 

Response 2: We agree with the reviewer that long-term follow-up studies are needed as this will be an important factor in care. With a limited cohort of patients and the retrospective nature of our study this was not one of our secondary aims.  

Comment 3: Absence of Control Group: Without a control group for comparison, it may be challenging to determine the effectiveness of mechanical thrombectomy compared to standard treatments like anticoagulation alone. 

Response 3: We had thought about using patients from our own institution initially, however we wanted to limit as much bias at a single institution as possible. With that in mind we decided to use reference 29 (large meta-analysis with 7,918 patients treated with anticoagulation alone) as our comparative group instead. 

Comment 4: Potential Selection Bias: The retrospective design of the study might introduce selection bias, as the patients included may not accurately represent the broader population with submassive PE, potentially compromising the validity of the findings. 

Response 4: We agree with the above and have changed the limitations section in our manuscript accordingly. 

Round 2

Reviewer 1 Report

Comments and Suggestions for Authors

Manuscript has been considerably improved post revision. I recommend acceptance of the manuscript in its current form.

Comments on the Quality of English Language

There are no major issues with quality of english. Some minor revisions might be done to improve readability.